# Boosting Resilience Attentional Bias in Previously Bullied University Students with Low Post-Traumatic Growth: A Transcranial Direct Current Stimulation Study

**DOI:** 10.3390/brainsci14111069

**Published:** 2024-10-27

**Authors:** Yennifer Ravelo, Rosaura Gonzalez-Mendez, Olga M. Alegre de la Rosa, Hipólito Marrero

**Affiliations:** 1Instituto Universitario de Neurociencias (IUNE), Universidad de La Laguna, 38200 La Laguna, Spain; yravelog@ull.edu.es (Y.R.); mrglez@ull.edu.es (R.G.-M.); oalegre@ull.edu.es (O.M.A.d.l.R.); 2Departamento de Psicología Cognitiva, Social y Organizacional, Universidad de La Laguna, 38200 La Laguna, Spain; 3Departamento de Didáctica e Investigación Educativa, Universidad de La Laguna, 38200 La Laguna, Spain

**Keywords:** emotional Stroop task, attentional bias, post-traumatic growth, transcranial direct current stimulation, intentionality, mentalizing network, approach motivation, bullying, university students

## Abstract

**Background/Objectives:** Post-traumatic growth (PTG) has the potential to draw positive consequences from trauma. Hence, there is interest in finding ways to promote PTG. Research has identified an attentional bias towards positive resilience-related words (e.g., “persistence”, “purpose”) in university students who report high PTG after experiencing adversities. Although people can respond to these experiences by showing low PTG, this bias seems to help with their struggle by making purposeful contents more accessible. Therefore, boosting attentional bias towards positive resilience-related words could help people with low PTG. **Methods:** In this study, the participants were thirty-six university students who had experienced bullying before entering university. Using a Stroop emotional task, they identified the color of resilience and neutral words, either positive or negative, before and after being submitted to transcranial direct current stimulation. Stimulation was targeted at the right temporal area involved in intentionality processing. **Results:** In the anodal condition, the results support a stimulation effect on the resilience attentional bias that could benefit participants with low PTG. A significant moderation of approach motivation for this effect was also found. Specifically, only when participants had medium or high approach motivation did stimulation boost the attentional bias in students with low PTG. **Conclusions:** These results support that tDCS stimulation in this brain area is effective in enhancing resilience attentional bias in low-PTG students. However, for this effect to occur it is necessary to have approach motivation, which is motivation related to goals.

## 1. Introduction

People may respond in different adaptive ways when faced with highly stressful events. In this way, they may maintain healthy levels of functioning, recover better after a temporary decline, or even thrive above previous performance [1,2,3]. Within this range of resilient responses, thriving is of particular interest to researchers because it has the potential to draw positive consequences from trauma. As defined by Tedeschi and Calhoun [4,5], post-traumatic growth (PTG) involves enduring positive changes that occur as a result of struggling to cope with a significant life challenge. Those changes are reported as strengthening relationships, a greater sense of personal strengths, a greater appreciation for life, new possibilities for one’s life, or spiritual development [5].

PTG has been found to be protective against the aftermaths of different traumatic experiences [2,6,7], although it may coexist with distress [8,9]. Bullying is a potential traumatic experience that can cause long-lasting negative consequences on health and academic performance [10,11]. However, there is also evidence indicating that it can lead to thriving [7,12,13]. PTG thus provides an avenue to intervene with people who have experienced trauma [14,15]. Hence, there is an interest in finding ways to promote thriving, such as those provided by cognitive neuroscience. In this regard, it is worth mentioning the role that attentional bias towards words and concepts plays in regulating behavior.

Research has shown diverse negative attentional biases in people with psychological maladjustment. For instance, anxious people have been found to pay more attention to threatening stimuli (images or words) in some tasks [16,17]. Moreover, depressed and addicted people seem especially attentive to stimuli related to their respective conditions [18]. To compensate for these un-adaptive attentional biases, interventions often involve training oneself to pay attention to appropriate stimuli or fostering attentional inhibition to wrong stimuli [19,20,21].

A different perspective involves looking at attentional biases as a source of potential benefits. Along these lines, a well-known attentional bias is the so-called positivity bias in the elderly, which favors positive over negative stimuli in cognitive processing [22]. This positivity effect seems to be driven by two different processes: an automatic attention bias toward positive stimuli and a controlled mechanism that diverts attention away from negative stimuli [23].

In a similar vein, research has found an attentional bias towards positive resilience-related words [24]. Based on normative studies, it was first confirmed that some words can be emotionally differentiated as positive (resilience facilitators) or negative (resilience inhibitors) depending on their semantic proximity to the concept of resilience. Thus, four lists of words were used in an emotional Stroop task to measure attentional latencies while participants identified the color of different words with emotionally relevant or neutral content. These four lists combined the valence (positive vs. negative) and the association with resilience (resilience-related vs. non-resilience-related) of the words. In this way, Gonzalez-Mendez et al. [24] found a main factor of resilience according to which participants took more time to identify resilience than non-resilience words. Additionally, there was a valence effect that meant that identifying the color took more time for positive than negative words. These results support that the resilient content of words is emotionally processed, as latency responses increase when attention is attracted during an emotional Stroop task. Moreover, participants who have suffered adversity recently and reported themselves as high in PTG paid more attention to positive resilience-related words in contrast to negative resilience-related words, whereas there was no difference in low-PTG participants. That is, more attention to positive than negative resilience words is associated with high PTG when struggling to overcome adversity. All this suggests that attentional bias towards positive resilience-related words might be useful in assisting the PTG process.

As defined by Todd and colleagues, affective attentional bias is “the predisposition to attend to certain categories of affectively salient stimuli over others” ([25], p. 365). This affective filtering process modulates emotional responses in a proactive rather than reactive manner [25]. If this is kept in mind, it can be hypothesized that the increase in a positive resilience attentional bias could help in the PTG process, as preference towards positive resilience-related words could make related concepts more accessible to the mind and thus be able to influence behavior in daily life. Therefore, we are interested in investigating the stimulation of target brain areas as a way to boost bias towards positive resilience-related words in people who report low PTG. Specifically, previous research using neuroimage and other neuroscience techniques has shown that the superior temporal sulcus (STS) is part of the mentalizing network [26] recruited for processing intentionality associated with goals [27,28,29,30,31,32,33] and social information [34], and it is usually stronger in the right hemisphere [35]. This makes this area particularly suitable for examining whether brain stimulation of it would increase attention towards words associated with intentionality. This is the case of positive resilience words associated with proactivity to overcome adversity, particularly in university students who reported low PTG after having suffered bullying in secondary school. Transcranial direct current stimulation (tDCS) is a non-invasive brain stimulation tool that has shown great potential in improving cognitive performance. Studies highlight tDCS’s role in elucidating cortical substrates that underlie cognitive functions [36]. tDCS uses mild and constant electrical currents (typically up to 2 mA) to induce short-term changes in the excitability and cortical activation of the brain regions. Depending on current polarity, it can either excite or inhibit activity. Anodic tDCS increases the probability of firing action potentials via neuronal membrane depolarization, enhancing spontaneous activity in the targeted region and consequently functionally connected areas [37]. This highlights a causal relationship between cognitive functions and underlying cortical structures. In fact, tDCS has been used to examine the effect of brain stimulation on attentional biases [38,39,40].

Previous research has also shown that greater attention to positive resilience-related words is associated with approach motivation [24]. In this regard, we are interested in examining whether the hypothesized effect of brain stimulation in boosting this resilience attentional bias is moderated by approach motivation. Specifically, we consider that approach motivation implies a willingness to overcome adversity, which could contribute to boosting the effect of stimulation on the resilience attentional bias in those students low in PTG.

### The Present Study

This study analyzes the modulation of PTG through excitatory (anodal) tDCS in the medial aspects of the right superior temporal sulcus (rSTS) on the attentional bias towards resilience-related words. The sample consists of university students who had experienced bullying before entering university. In particular, we are interested in students who could still be struggling to overcome adversity, given that they reported low PTG. In this study, we explored whether tDCS in the target region of interest (ROI) induces a positive resilience attentional bias in those who scored lower in terms of PTG, which could assist them in the process of struggling with the experience of bullying. Specifically, the hypotheses are as follows.

**Hypothesis** **1.**
*The effect of stimulation on resilience attentional bias is associated with PTG, such that the lower the PTG, the greater the attentional bias after stimulation.*


**Hypothesis** **2.**
*The effect of stimulation on the resilience attentional bias associated with low PTG is moderated by approach motivation.*


## 2. Methods

### 2.1. Participants

Participants were selected from an initial sample of 133 undergraduate students of different degrees at the Universidad de La Laguna (La Laguna, Spain). They had all experienced bullying before entering university, which often occurs because they exhibit characteristics distinct from their peers, such as academic performance, cultural background, gender, ethnicity, religion, or any other. Of all of them, 36 agreed to participate in the experiment (83.3% were women and 16.7% were men). The ages ranged from 18 to 23. The average age was 21 (*SD* = 5.75). Exclusion criteria were suffering from epilepsy (or having close relatives affected), migraine, brain damage, cardiac, neurological or psychiatric disease, having any injury, or subcutaneous metal in any of the two parts where electrodes would be placed. While 18 participants were randomly assigned to the anodal stimulation condition, the other 18 were allocated to the sham (placebo) condition. 

### 2.2. Materials and Stimuli

#### 2.2.1. Post-Traumatic Growth

Post-traumatic growth was measured using the 9-item scale from the Resilience Portfolio Measurement Packet [41] (e.g., “I established a new path for my life”, “I am able to do better things with my life”). This scale was selected due to the availability of its Spanish translation, which has shown strong psychometric properties and predictive validity [42]. Response options ranged from 1 (not true) to 4 (mostly true). Cronbach’s alpha was 0.88.

#### 2.2.2. BAS Approach Motivation Trait

The BIS/BAS scale [43] assesses both the Behavioral Inhibition System (BIS) and the Behavioral Activation System (BAS). The BAS is involved in regulating appetitive motivations that direct behavior towards desirable outcomes, whereas the BIS regulates aversive motivations that aim to avoid negative or unpleasant stimuli. This scale has a Spanish version whose psychometric properties are comparable to those of the English version [44]. In this study, we only used the BAS subscale, which consists of thirteen items (e.g., “If I see a chance to get something I want, I move on it right away”). Response options range from 1 (strong disagreement) to 4 (strong agreement). Cronbach’s alpha was 0.60.

#### 2.2.3. Emotional Stroop Task

The emotional Stroop task used in this study was developed by Gonzalez-Mendez et al. [24]. It consists of 48 words grouped into 4 conditions: (1) positive resilience (e.g., optimism); negative resilience (e.g., pessimism); positive non-resilience (e.g., elegance); and negative non-resilience (e.g., arrogance). Arousal and psycholinguistic factors, such as frequency and syllabic and letter length of the words, were balanced using the Espal database [45].

The task consisted of identifying the color of each of the words, which were presented separately on a screen. Specifically, the participants had to press the key on the keyboard associated with each color (red, blue, green, or yellow). The four colors were associated in a counterbalanced way with the words from each of the four lists (see Table 1).

### 2.3. Design and Procedure

A pre-post tDCS stimulation design was adopted. The dependent measure was an attentional index towards positive resilience words in the Stroop task. The dependent measure was taken before and after stimulation. Participants were submitted to anodal or sham (placebo) stimulation. The presence of a placebo (sham) condition was not disclosed to ensure participants remained unaware of the specific tDCS condition they were undergoing.

The design of this study was made in compliance with the Declaration of Helsinki and approved by the Institutional Ethics Committee of the University of La Laguna (CEIBA2022-3216). A questionnaire that collected information about the students’ adversities and their level of PTG was initially filled out by first-year undergraduate students. After identifying those who had suffered bullying before entering the college (*n* = 133), they were invited to the laboratory to be informed of the general objective of the current study. Those students who agreed to participate completed a personal data form, a screening questionnaire for the potential exclusion criteria, and signed a consent form. None of them reported having epilepsy (or having relatives affected), brain damage, migraines, heart disease, or other psychological or medical conditions, and all of them were right-handed according to the Edinburgh Handedness Inventory [46]. The anonymity and confidentiality of the data were guaranteed at all times.

The participants sat in front of a laptop with a Linux operating system in which the experiment was programmed in PsychoPy2 1.83.01 [47]. At the beginning of this study, they were given a practical trial with eight words. Once familiarized with how to respond, they performed the emotional Stroop task at two different times, before stimulation and again after stimulation. The participants were asked to respond by indicating the color of each word by pressing the key on the keyboard associated with each color (red color -> letter “h”; blue color -> “j”; green color -> letter “k”; and yellow color -> “l”). Stimuli in each list were shown in a random order, and their colors were counterbalanced in the four lists.

### 2.4. Transcranial Direct Current Stimulation Protocol

Transcranial direct current stimulation (tDCS) is a non-invasive brain stimulation tool that has shown great potential in improving cognitive performance. Studies highlight tDCS’s role in highlighting cortical substrates that underlie cognitive functions [36]. tDCS uses mild and constant electrical currents (typically up to 2 mA) to induce short-term changes in the excitability and cortical activation of regions of the brain. Depending on current polarity, it can either excite or inhibit activity. Anodic tDCS increases the probability of firing action potentials via neuronal membrane depolarization, enhancing spontaneous activity in the targeted region and consequently functionally connected areas [37]. This demonstrates a causal relationship between cognitive functions and underlying cortical structures.

In this study, we used a CE-certified battery-driven electrical stimulator (TCT Research Ltd., Hong Kong, China) with an intensity of 2 mA for 20 min, plus 20 s fade-in and fade-out phases. The stimulation parameters were considered safe [48]. We used rubber electrodes sized 5 cm × 5 cm and 7 cm × 5 cm, covered saline-soaked sponges, yielding a density of 0.08 mA/cm^2^ and 0.057 mA/cm^2^, respectively. The smaller electrode was placed in the T8 area according to the International System 10/20, which aligns with the rSTS, while the cathodal electrode was placed extracranially in the contralateral shoulder to minimize the effects on the brain and improve the size effect of active tDCS [49]. As shown in Figure 1, the tDCS configuration targeted the rSTS that is supported by the SimNIBS 4 (Simulation of Non-invasive Brain Stimulation) software package [50]. The stimulation time was established based on previous studies of tDCS [30,51,52]. The sham tDCS stimulation followed the same procedure as in the anodic stimulation, with the same electrode setup. The only difference was that, in the sham condition, the fade-in and fade-out phases lasted for 20 s and 3 min of the stimulation.

### 2.5. tDCS Procedure

Once the participants completed the emotional Stroop task (before stimulation), they were fitted with electrodes following the 20 min tDCS protocol of anodal or placebo stimulation (sham condition). After the tDCS, equipment was removed. Then, the participants performed the task again. The entire session lasted approximately 40 min.

At the end of the experimental session once the task was completed, the participants were asked to report discomfort or any adverse side effects (see Table 2) during tDCS [53,54]. Finally, they were thanked for their collaboration and given a brief explanation of the study.

### 2.6. Statistical Analyses

Given that affective attentional bias is characterized by the predisposition to attend to certain categories of affectively salient stimuli over others, resilience attentional bias was measured using an index, which was calculated on latencies as follows: (resilience positive words minus non-resilience positive words) − (resilience negative words minus non-resilience negative words). In this sense, the more positive the index, the greater the bias towards positive resilience-related words. The Kolmogorow–Smirnoff test showed that attentional bias pre-stimulation, attentional bias post-stimulation, and PTG were over the probability of 0.05 in the whole sample, and also in the anodal and sham groups, which supports distribution normality.

To test the hypothesis that increased resilience attentional bias after active stimulation is associated with low PTG, we first performed Pearson correlations between the reported PTG and the resilience attentional bias both pre- and post-stimulation. Concurrently, the Fisher-z transformation method, as suggested by Eid et al. [55], was applied to compare correlation strengths from dependent samples in order to evaluate whether the correlation with “attentional bias-PTG” significantly changed from “pre” to “post” stimulation in both the anodal and the sham groups. To test for the specificity of the hypothesized associations, we used methods detailed by Lenhard and Lenhard [56]. The comparison was made using the online calculator available at https://www.psychometrica.de/correlation.html#independent accessed on 25 September 2024 [56].

To test Hypothesis 2, a simple moderating effect of approach motivation was tested using the SPSS26 macro program PROCESS 4.1 [57]. Specifically, Model 1 was computed. The bootstrap method was used in the mediation analysis to calculate the 95% confidence intervals for each of the 10,000 repeated samples. Statistical support for the moderation was assumed when zero was outside of the confidence interval.

## 3. Results

### 3.1. Comparison of Correlations from Dependent Samples

Separately for the anodal and sham groups, Table 3 shows the Pearson correlations between PTG and resilience attentional bias for pre- and post-tDCS stimulation conditions. As can be seen, there is a significant negative correlation between PTG and the attentional bias post-stimulation (in bold), indicating a greater attentional bias in those with lower PTG. In addition, the Z test revealed that the correlation between PTG and attentional bias significantly changed from pre-stimulation to post-stimulation, but only in the Anodal group. This result supports Hypothesis 1, i.e., the participants with low PTG show a greater resilience attentional bias after stimulation.

### 3.2. Moderating Role of BAS on the Link Between PTG and Resilience Attentional Bias After Stimulation

Two simple moderation models were computed to test Hypothesis 2. Specifically, approach motivation was used as a moderating variable between PTG and the resilience attentional bias. This analysis was repeated for each of the experimental conditions (anodal and sham). As expected, approach motivation plays a moderating role in the anodal condition, but not in the sham condition (see Table 4).

No significant direct association was found between PTG and post-stimulation attentional bias, but approach motivation moderated the association between both factors (Figure 2). As indicated by the conditional effects of PTG at different values of approach motivation (see Table 4), low-PTG participants only benefited from stimulation when they obtained a medium score, and especially a high score, on approach motivation. The percentage of variance of attentional bias explained by the model condition is high (61%).

## 4. Discussion

While some individuals maintain low PTG scores after adversity, others report experiencing thriving [58,59]. This has led to interest in finding ways to promote PTG. Research has identified an attentional bias towards positive resilience-related words (e.g., “persistence”, “purpose”) in people who report high PTG, which could help their struggle with adversity by making purposeful contents more accessible [24]. Based on this finding, the current study aims to examine whether tDCS of the rSTS, involved in intentionality processing, boosts resilience attentional bias in university students who report low PTG after experiencing bullying. Using a Stroop emotional task, the participants identified the color of resilient and neutral words, either positive or negative, before and after being subjected to tDCS. A positive resilience attentional index was computed using the latencies for further analyses.

The comparison of correlations between PTG and pre- and post-attentional bias only showed significant differences in the anodal condition. Furthermore, only the students who received stimulation and were low in PTG showed an increase in their resilience attentional bias, which supports H1. Moderation analysis also supported H2 by showing the moderating role of approach motivation. Specifically, lower PTG was associated with greater attentional bias, but only when participants scored medium, and especially high, on approach motivation.

The rSTS is part of the mentalizing network involved in processing intentionality, and the results of this study support its role in processing words associated with resilience goals and intentionality. Despite the small sample size, rSTS thus emerges as a brain area suitable for cognitive interventions aimed at improving behavioral regulation in response to adversity and trauma. Keeping these concepts active in mind is expected to help more adaptive behavioral regulation in people with low PTG. However, further research is required to develop effective brain stimulation interventions that boost resilience attentional bias. For example, it is necessary to examine the number of tDCS sessions that would be needed, as well as assessing their effect using indicators of psychological functioning and well-being by contrasting participants in the anodal and placebo conditions. Additionally, it would be necessary to examine whether tDCS brain stimulation can be combined with cognitive training. Attentional training to positive resilience words could be used by self-instruction to adopt these words, or even by presenting participants with situations representing everyday examples of proactive responses to adversity associated with concepts such as persistence or purpose.

In the moderation analysis, the interaction found indicates that the association between cognitive stimulation and resilience attentional bias is moderated by motivation. Basically, the positive effect of cognitive stimulation is more pronounced when the motivational factor is present or stronger. Previous approach motivation thus seems to act as an attentive predisposition towards facilitator stimuli for struggling with adversity, such as positive resilience-related words, which may be boosted by stimulation of the brain’s intentionality network. Consistent with this interpretation, Zhou et al. [59] found that adolescents high in PTG following the Wenchuan earthquake showed more hyperarousal symptoms and fewer symptoms of avoidance, which could indicate an attentive predisposition to search for ways to cope. These findings should then be considered in future interventions, since brain stimulation itself does not seem to facilitate greater attention to resilient words in low-PTG students when they score low on approach motivation.

### Limitations and Their Implications for Future Research

There are several limitations in this study that require consideration. First, transcranial direct current stimulation (tDCS) has inherent limitations. In this study, the right superior temporal sulcus (rSTS) was stimulated using the 10/20 EEG system for electrode placement. However, anatomical variability between participants and potential reductions in the focality of the stimulation protocol may have influenced the results. Future studies should consider incorporating techniques such as functional MRI (fMRI) or transcranial magnetic stimulation (TMS) to provide more direct and precise evidence of rSTS recruitment and stimulation effects, particularly in relation to resilience attentional bias.

Another limitation is the small sample size, which reduces the statistical power of the analysis. In addition, the sample predominantly comprised young female students, which limits the generalizability of the findings. Future research should be aimed at replicating the results with a larger and more gender-balanced sample. Increasing the number of stimuli in the Stroop Task could also enhance the statistical power and robustness of the findings.

Moreover, this study has focused on a single type of adverse experience. Therefore, it is necessary to verify the findings in people exposed to forms of adversity other than bullying. Research should also assess the effectiveness of non-invasive brain stimulation interventions, not just within university populations but across more diverse groups as well.

To the best of our knowledge, this is the first study to examine the effect of tDCS on a specific brain area to boost attentional bias towards positive resilience words. The results support that this bias is boosted by tDCS anodal stimulation of the rSTS in low-PTG students when they score medium or high in approach motivation. However, it is still necessary to examine whether this increase contributes to improvements in psychological functioning. If confirmed, this would open up an avenue to possible interventions that combine tDCS with other strategies to promote PTG after bullying or another adverse experience. Menesini [60] has highlighted the need to identify the participants who are most likely to benefit from anti-bullying interventions, as well as the factors that moderate the effectiveness of those interventions. In this sense, our findings point to approach motivation as a moderating factor to be considered.

## Figures and Tables

**Figure 1 brainsci-14-01069-f001:**
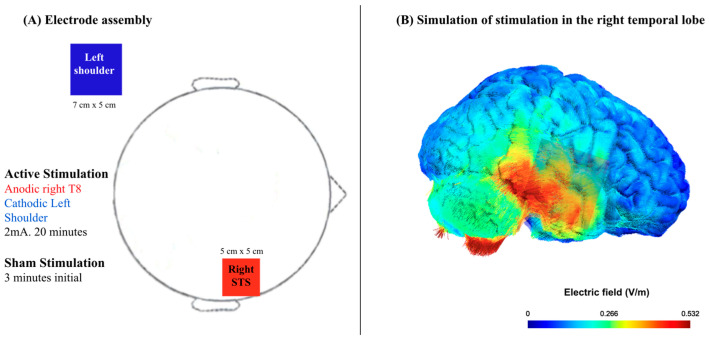
Simulation of electric field strength by SimNIBS 4 (Simulation of Non-invasive Brain Stimulation) of the active (anodal) electrode in the right superior temporal sulcus (STS). (**A**) Electrode assembly and (**B**) simulation of stimulation in the right STS.

**Figure 2 brainsci-14-01069-f002:**
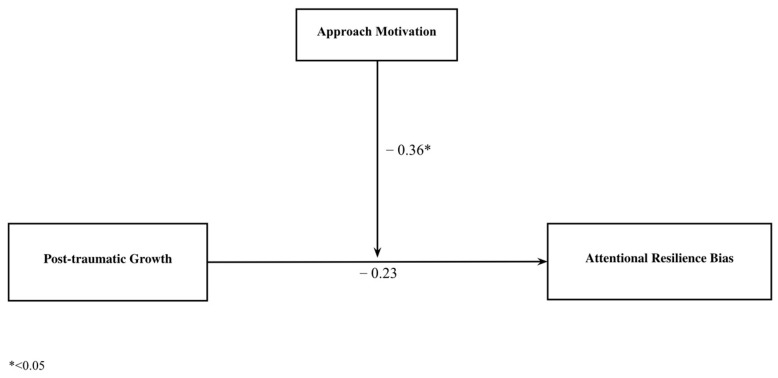
Moderating effect of approach motivation on the relationship between post-traumatic growth and resilience attentional bias in the anodal condition.

**Table 1 brainsci-14-01069-t001:** List of words in the emotional Stroop task.

Training Words	Resilience Positive	Resilience Negative	Non-Resilience Positive	Non-Resilience Negative
Earth	Coping	Abandonment	Beauty	Boredom
Matter	Hope	Apathy	Fun	Arrogance
Space	Firmness	Exhaustion	Cordiality	Arrogance
Landscape	Strength	Cowardice	Elegance	Brutality
Structure	Persistence	Defeat	Idealism	Ugliness
Speed	Optimism	Weakness	Equality	Stupidity
Trade	Endurance	Pessimism	Nobility	Slowness
Episode	Purpose	Fragility	Cleaning	Rudeness
	Serenity	Resignation	Originality	Rigidity
	Overcoming	Renounce	Punctuality	Dirtiness
	Animosity	Impotence	Smoothness	Vulgarity
	Courage	Submission	Simplicity	Clumsiness

**Table 2 brainsci-14-01069-t002:** Adverse effects, severity, and rounded percentage of participants that experienced them in the tDCS study.

Type of Effect	Severity	%Anodal	%Sham
Headache	Mild	11.10%	16.70%
Neck pain	Mild	5.60%	5.60%
Scalp pain	Moderate	5.60%	0%
Tingling	Moderate	22.20%	16.70%
Itching	Moderate	16.70%	16.70%
Hot sensation	Mild	22.20%	11.10%
Reddening of the skin	Mild	16.70%	11.10%
Drowsiness	Mild	22.20%	22.20%
Concentration problems	Mild	11.10%	5.60%
Acute mood change	Mild	5.60%	0%

**Table 3 brainsci-14-01069-t003:** Pearson correlations between PTG and attentional bias pre- and post-stimulation, separately for anodal and sham groups, and significance level when comparing before and after stimulation correlations using Z test.

Variables	ANODAL		SHAM	
	BIAS Post	PTG	BIAS Post	PTG
BIAS pre	−0.392 (*p* = 0.108)	0.116 (*p* = 0.648)	0.257 (*p* = 0.302)	−0.337 (*p* = 0.172)
BIAS post	1	−0.567 (*p* = 0.014)	1	−0.257 (*p* = 0.303)
Z test		1.751		−0.271
Z test probability		0.04		0.393

**Table 4 brainsci-14-01069-t004:** Regression coefficients and confidence intervals in two simple moderation models, computed separately for anodal and sham conditions.

Anodal Condition									
Outcome Variable	Predictor Variable	COEFF	*SE*	*t*	LLCI	ULCI	*p*	*R* ^2^	*F*
Attentional bias	PTG	−0.01	0.13	−1.6	−0.54	0.08	0.13	0.61	7.26
	BAS	−0.03	0.13	−0.16	0.37	−0.32	0.87		
	Int_1	−0.36	0.15	−2.46	−0.68	−0.05	0.03		
Conditional effects of PTG at values of approach motivation as moderator									
	−0.72	0.03	−0.21	0.15	−0.41	0.47	0.88		
	0.19	−0.29	0.13	−2.2	−0.59	−0.01	0.04		
	0.65	−0.46	0.14	−3.31	−0.76	−0.16	0		
Sham condition									
Attentional bias	PTG	−0.57	0.3	−1.9	−1.22	0.07I	0.07	0.29	1.89
	BAS	0.54	0.29	1.84	−0.09	1.16	0.09		
	Int_1	−0.35	0.26	−1.32	−0.92	0.21	0.21		

LLCI, lower level of the 95% confidence interval; ULCI, upper level of the 95% confidence interval; *SE*, Standard Error.

## Data Availability

The data presented in this study are available on request from the corresponding author due to ethical reasons.

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
