# Peer review of "Boosting Resilience Attentional Bias in Previously Bullied University Students with Low Post-Traumatic Growth: A Transcranial Direct Current Stimulation Study"

_brainsci, 2024, doi:10.3390/brainsci14111069_

Round 1

Reviewer 1 Report

Comments and Suggestions for Authors

Thanks a lot for providing me with the opportunity to review the manuscript. I hope you find the following comments useful to improve the quality of your valuable paper.

Introduction

1. In general, the introduction is too long, focusing mainly on the psychological aspects of trauma. As a study on tDCS, it is suggested to add more information about the neurocognitive function of brain, neuroimaging studies, and also introducing the tDCS and related studies investigating the effect of tDCS or other NIBS techniques on “Resilience Attentional Bias”.

2. Line 44-46: "In this regard, it is worth mentioning the role that attentional bias towards words and concepts can exert in regulating behavior aimed at struggling with adversity."

The phrase "can exert in regulating behavior aimed at struggling with adversity" is somewhat convoluted. Simplifying it could enhance clarity.

Method

1. I could not find any paragraph presenting a whole picture on procedure, type of study, and blinding.

2. Please provide more information about the pyschometric properties of the “ 9-item scale from the Resilience Port- 130 folio Measurement Packet”  and  “Behavioral Activation Systemand cite related references.

3. Did you assess the diversity?

4. How did you score the “Emotional Stroop task 

5. Has any ethics committee approved your study?

6. Please provide more information about the brand, company and country that the tDCS device has been manufactured?

7. What do you mean by “extracranially” in line 191?

8. How long was the duration of stimulation in sham tDCS?

9. In statistical analyziz section, It is not clear how the effect of tDCS on attention bias has been assessed?

Author Response

Comments 1: In general, the introduction is too long, focusing mainly on the psychological aspects of trauma. As a study on tDCS, it is suggested to add more information about the neurocognitive function of brain, neuroimaging studies, and also introducing the tDCS and related studies investigating the effect of tDCS or other NIBS techniques on “Resilience Attentional Bias”.

Response 1: We agree that Introduction seems too long. However, we consider that making it shorter would leave important aspects unclear. In fact, reviewer 2 has asked us to expand on some content.

Regarding the first suggestion, we have included the following text in the Introduction section, on line 85 about Transcranial direct current stimulation (tDCS), and its use in research about attentional biases:

“Transcranial direct current stimulation (tDCS) is a non-invasive brain stimulation tool that has shown great potential in improving cognitive performance. Studies highlight tDCS’s role in elucidating cortical substrates that underlie cognitive functions [36]. tDCS uses mild and constant electrical currents (typically up to 2mA) to induce short-term changes in the excitability and cortical activation of the brain regions. Depending on current polarity, it can either excite or inhibit activity. Anodic tDCS increases the probability of firing action potentials via neuronal membrane depolarization, enhancing spontaneous activity in the targeted region and consequently functionally connected areas [37]. This highlights a causal relationship between cognitive functions and underlying cortical structures. In fact, tDCS has been used to examine the effect of brain stimulation on attentional biases [38,39,40]”.

We also added these references in line 98 to previous research of tDCS effects on attentional biases:

[38] Smits, F.M.; Schutter, D.J.; van Honk, J.; Geuze, E. Does non-invasive brain stimulation modulate emotional stress reactivity? Social cognitive and affective neuroscience 2020,15, 23–51.

[39] Sanchez-Lopez, A.; De Raedt, R.; Puttevils, L.; Koster, E.H.; Baeken, C.; Vanderhasselt, M. Combined effects of tDCS over the left DLPFC and gaze-contingent training on attention mechanisms of emotion regulation in low-resilient individuals. Prog Neuro-Psychopharmacol Biol Psychiatry 2021,108.

Finally, in lines 83-84 we have specified that specialization of rSTS in intentionality processing is supported by neuroimaging studies as a part of the mentalizing network with the references.

Comments 2: Line 44-46: "In this regard, it is worth mentioning the role that attentional bias towards words and concepts can exert in regulating behavior aimed at struggling with adversity." The phrase "can exert in regulating behavior aimed at struggling with adversity" is somewhat convoluted. Simplifying it could enhance clarity.

Response 2: This sentence has been simplified and now read as follows:

“In this regard, it is worth mentioning the role that attentional bias towards words and concepts plays in regulating behavior”.

Comments 3: I could not find any paragraph presenting a whole picture on procedure, type of study, and blinding.

Response 3: Thanks for this comment. The following paragraph was added to in lines 170-175:

Design and procedure

A pre-post tDCS stimulation design was adopted. The dependent measure was an attentional index towards positive resilience words in the Stroop task. The dependent measure was taken before and after stimulation. Participants were submitted to anodal or sham (placebo) stimulation. The presence of a placebo (sham) condition was not disclosed to ensure participants remained unaware of the specific tDCS condition they were undergoing”.

Comments 4: Please provide more information about the psychometric properties of the “ 9-item scale from the Resilience Port-folio Measurement Packet”  and  “Behavioral Activation System” and cite related references.

Response 4: In the materials and stimuli section, we have expanded the information on the psychometric properties of each of these scales.

(1) With respect to the 9-item scale from the Resilience Port-folio Measurement Packet, it now reads as follows:

“This scale was selected due to the availability of its Spanish translation, which has shown strong psychometric properties and predictive validity [42]”.

In addition, we have added the following reference:

[42] Gonzalez-Mendez, R.; Ramírez-Santana, G.; Hamby, S. Analyzing Spanish adolescents through the lens of the Resilience Portfolio Model. J Interpers Violence 2021, 36, 4472–4489.

(2) With respect to the The BIS/BAS scale, it now reads as follows:

“This scale has a Spanish version whose psychometric properties are comparable to those of the English version [44]”.

In addition, we have added the following reference:

[44] Perczek, R.; Carver, C.S.; Price, A.A.; Pozo-Kaderman, C. Coping, mood, and aspects of personality in Spanish translation and evidence of convergence with English versions. J Pers Assess 2000, 74, 63–87.

Comments 5: Did you assess the diversity? 

Response 5: All participants of this study had experienced bullying, which often occurs because they exhibit characteristics distinct from their peers, such as academic performance, cultural background, gender, ethnicity, religion, or any other. They were selected from an initial sample of 133 students of different grades, which guarantees greater diversity. The university's diversity is also growing, thanks to a rising migrant population, students from other European countries via the Erasmus program, and explicit university policies supporting diversity across gender, and physical/ intellectual disabilities.

The following paragraph has been modified in the participant section to clarify this issue: “Participants were selected from an initial sample of 133 undergraduate students of different degrees at the Universidad de La Laguna (La Laguna, Spain). They had all experienced bullying before entering university, which often occurs because they exhibit characteristics distinct from their peers, such as academic performance, cultural background, gender, ethnicity, religion, or any other. Of all of them, 36 agreed to participate in the experiment”.

Comments 6: How did you score the “Emotional Stroop task”

Response 6: The emotional Stroop task takes latencies of naming the color of the words with  emotional content as an index of emotionality attracting attention. As it is described in the Statistical analysis section: “Given that affective attentional bias is characterized by the predisposition to attend to certain categories of affectively salient stimuli over others, resilience attentional bias was measured using an index, which was calculated on latencies as follows: (resilience positive words minus non-resilience positive words) – (resilience negative words minus non-resilience negative words). In this sense, the more positive the index, the greater the bias towards positive resilience-related words”.

Comments 7: Has any ethics committee approved your study?

Response 7: The following paragraph has been included in the Procedure section:

“The design of the study was conducted in compliance with the Declaration of Helsinki, and approved by the Institutional Ethics Committee of University of La Laguna (CEIBA2022-3216)”.

Comments 8: Please provide more information about the brand, company and country that the tDCS device has been manufactured?

Response 8: We have included the following information about the brand in the manuscript:

“(TCT Research Ltd., Hong Kong, China)”.

Comments 9: What do you mean by “extracranially” in line 191?

Response 9: We have now expanded the information in the Transcranial direct current stimulation protocol section:

“The cathodal electrode was placed extracranially in the contralateral shoulder to minimize the effects on the brain and”.

Comments 10: How long was the duration of stimulation in sham tDCS?

Response 10: We have clarified how many minutes the stimulation lasted in tDCS.  This information has been included in the Transcranial direct current stimulation protocol section:

“...and 3 minutes of stimulation.”

Comments 11: In the statistical analysis section, It is not clear how the effect of tDCS on attention bias has been assessed?

Response 11: We carried out contrasts between the correlation of the attentional index pre and post stimulation. This type of statistical contrasts allowed us to test whether the association between the attentional index and PTG had significantly changed after stimulation: “Eid, M.; Gollwitzer, M.; Schmitt, M. Statistik und Forschungsmethoden Lehrbuch; Beltz: Weinheim, Germany, 2011”. Likewise, this analysis strategy based on correlations also facilitated carrying out a moderation analysis using PROCESS: “Hayes, F. Process Procedure for SPSS, 3rd ed.; New York: Guilford Press, 2022”, a statistical tool that is gaining great influence in this type of studies.

Reviewer 2 Report

Comments and Suggestions for Authors

The study explores the effects of tDCS in boosting resilience in thirty-six students who had experienced bullying before entering the university. The topic is interesting and I believe the use of TDCS in psychology deserves attention, however, there are several issues that should be addressed before considering the manuscript for publication.

-            Firstly, the meaning of post traumatic growth should be introduced and more references to the current literature added. So I suggest to strengthen the introduction

-            Similarly, tDCS should be introduced, which literature suggests that tDCS can be the elective tool to explore the hypothesis of the current study? Are there studies applying tDCS in similar psychological constructs? In attention? What tDCS is? I think this is important for the background of the study. See for instance a recent metanalysis reporting the effects of tDCS in cognitive performances from Giustiniani et al., 2024

Giustiniani, A., Maistrello, L., Mologni, V., Danesin, L., & Burgio, F. (2024). TMS and tDCS as potential tools for the treatment of cognitive deficits in Parkinson’s disease: a meta-analysis. Neurological Sciences, 1-14.

-            It’s not clear where the 133 undergraduate students come from and how they were enrolled

-            It’s not clear how participants were assigned to each group

-            Paragraph 2.4 is missing many references when describing the tDCS

-            Why did the authors decide to not perform a repeated measure ANOVA? I’m not sure about the use of the statistics reported in the paper;

-            The role of bullying is barely mentioned throughout the manuscript. Instead, due to the title of the manuscript and the aims of the work, findings should be discussed in light of its role in the observed results.

Comments on the Quality of English Language

A minor editing can improve the manuscript

Author Response

Comments 1: Firstly, the meaning of post traumatic growth should be introduced and more references to the current literature added. So I suggest to strengthen the introduction.

Response 1: Given that reviewer 1 found the introduction too long, we have only added some information on positive changes associated with post traumatic growth. Specifically, we have included the following sentence and some new references:

“Those changes are reported as strengthening relationships, a greater sense of personal strengths, a greater appreciation for life, new possibilities for one’s life, or spiritual development [5]”.

In addition, the following references have been included:

[7] Ratcliff, J.J.; Lieberman L.; Miller, A.K.; Pace, B. Bullying as a source of posttraumatic growth in individuals with visual impairments. Journal of Developmental and Physical Disabilities 2017, 29, 265–278.

[9] Hamby S, Taylor E, Segura A, Weber M. A dual-factor model of posttraumatic responses: Which is better, high posttraumatic growth or low symptoms? Psychological trauma: theory, research, practice, and policy 2022.

[14] Susanti, H.; Putri, A.F.; Susanti, S.S.; Malini, H.; Alim, S.; Bintari, D.R. Improving post-traumatic growth of disaster survivors: an integrative literature review. International emergency nursing 2024,75.  

Comments 2: Similarly, tDCS should be introduced, which literature suggests that tDCS can be the elective tool to explore the hypothesis of the current study? Are there studies applying tDCS in similar psychological constructs? In attention? What tDCS is? I think this is important for the background of the study. See for instance a recent metanalysis reporting the effects of tDCS in cognitive performances from Giustiniani et al., 2024.

Response 2: Thanks for your comment. We have included the following text in the Introduction section, on line 85 about Transcranial direct current stimulation (tDCS), and its use in research about attentional biases.

“Transcranial direct current stimulation (tDCS) is a non-invasive brain stimulation tool that has shown great potential in improving cognitive performance. Studies highlight tDCS’s role in elucidating cortical substrates that underlie cognitive functions [36]. tDCS uses mild and constant electrical currents (typically up to 2mA) to induce short-term changes in the excitability and cortical activation of the brain regions. Depending on current polarity, it can either excite or inhibit activity. Anodic tDCS increases the probability of firing action potentials via neuronal membrane depolarization, enhancing spontaneous activity in the targeted region and consequently functionally connected areas [37]. This highlights a causal relationship between cognitive functions and underlying cortical structures. In fact, tDCS has been used to examine the effect of brain stimulation on attentional biases [38,39,40]”.

The suggested reference has been included.

[38] Smits, F.M.; Schutter, D.J.; van Honk, J.; Geuze, E. Does non-invasive brain stimulation modulate emotional stress reactivity? Social cognitive and affective neuroscience 2020, 15, 23–51.

[39] Sanchez-Lopez, A.; De Raedt, R.; Puttevils, L.; Koster, E.H.W.; Baeken, C.; Vanderhasselt, M. Combined effects of tDCS over the left DLPFC and gaze-contingent training on attention mechanisms of emotion regulation in low-resilient individuals. Prog Neuro-Psychopharmacol Biol Psychiatry 2021, 108.

[40] Giustiniani, A.; Maistrello, L.; Mologni, V.; Danesin, L.; Burgio, F. TMS and tDCS as potential tools for the treatment of cognitive deficits in Parkinson’s disease: a meta-analysis. Neurological Sciences 2024, 1-14.

Comments 3: It’s not clear where the 133 undergraduate students come from and how they were enrolled.

Response 3: The Participants section has been rewritten to clarify this issue. The text now reads as follows:

“Participants were selected from an initial sample of 133 undergraduate students of different degrees at the Universidad de La Laguna (La Laguna, Spain). They had all experienced bullying before entering university, which often occurs because they exhibit characteristics distinct from their peers, such as academic performance, cultural background, gender, ethnicity, religion, or any other. Of all of them, 36 agreed to participate in the experiment”.

Comments 4: It’s not clear how participants were assigned to each group.

Response 4: Participants were randomly assigned to each group as it is described in “Participants”. 

Comments 5: Paragraph 2.4 is missing many references when describing the tDCS.

Response 5: We have checked that all references are included. In addition, we have added some new references to support the information in this section:

[30] Marrero, H.; Yagual, S.N.; García-Marco, E.; Gámez, E.; Beltrán, D.; Díaz, J.M.; Urrutia, M. Enhancing memory for relationship actions by transcranial direct current stimulation of the superior temporal sulcus. Brain Sciences 2020, 10, 497.

[52] García-Marco, E.; Nuez Trujillo, A.; Padrón, I.; Ravelo, Y.; Fu, Y.; Marrero, H. Negation and social avoidance in language recruits the right inferior frontal gyrus: a tDCS study. Front. Psychol. 2024, 15.

Comments 6: Why did the authors decide to not perform a repeated measure ANOVA? I’m not sure about the use of the statistics reported in the paper.

Response 6: We consider that correlational analysis was a good option, given the small size of our sample. We carried out contrasts between the correlation of the attentional index pre and post stimulation. This type of statistical contrasts allowed us to test whether the association between the attentional index and PTG had significantly changed after stimulation: “Eid, M.; Gollwitzer, M.; Schmitt, M. Statistik und Forschungsmethoden Lehrbuch; Beltz: Weinheim, Germany, 2011”.  Also important, this analysis strategy based on correlations facilitated carrying out a moderation analysis using PROCESS: “Hayes, F. Process Procedure for SPSS, 3rd ed.; New York: Guilford Press, 2022”, a statistical tool that is gaining great influence in this type of studies.

Comments 7: The role of bullying is barely mentioned throughout the manuscript. Instead, due to the title of the manuscript and the aims of the work, findings should be discussed in light of its role in the observed results.

Response 7: We have now included information about bullying in the introduction section:

“Bullying is a potential traumatic experience that can cause long-lasting negative consequences on health and academic performance [10, 11]. However, there is also evidence indicating that it can lead to thriving [7,12,13]”.

[7] Ratcliff, J.J.; Lieberman L.; Miller, A.K.; Pace, B. Bullying as a source of posttraumatic growth in individuals with visual impairments. Journal of Developmental and Physical Disabilities 2017, 29, 265–278.

[10] Laith, R.; Vaillancourt, T. The temporal sequence of bullying victimization, academic achievement, and school attendance: A review of the literature. Aggression and Violent Behavior 2022, 64.

[11] Li, C.; Wang, P.; Martin-Moratinos, M.; Bella-Fernandez, M.; Blasco-Fontecilla, H. Traditional bullying and cyberbullying in the digital age and its associated mental health problems in children and adolescents: a meta-analysis. Eur Child Adolesc Psychiatry 2024, 33, 2895–2909.

[12] Andreou, E.; Tsermentseli, S.; Anastasiou, O.; Kouklari, E.C. Retrospective accounts of bullying victimization at school: associations with post-traumatic stress disorder symptoms and post-traumatic growth among university students. Journal of Child & Adolescent Trauma, 2021, 14, 9–18.

[13] Ravelo, Y.; Alegre, O.M.; Marrero, H.; Gonzalez-Mendez, R. Motivational mediation between coping and post-traumatic growth in previously bullied college students. Front. Psychol. 2022, 13.

RESPONSE: Additionally, we have also added a paragraph at the end of the discussion:

“If confirmed, this would open an avenue to possible interventions that combine tDCS with other strategies to promote PTG after bullying or another adverse experience. Menesini [60] has also highlighted the need to identify the participants who are most likely to benefit from anti-bullying interventions, as well as the factors that moderate the effectiveness of those interventions. In this sense, our findings point to approach motivation as a moderating factor to be considered.”

[60] Menesini, E. Translating knowledge into interventions: An ‘individual by context’approach to bullying. European Journal of Developmental Psychology 2019, 16, 245–267.

Round 2

Reviewer 2 Report

Comments and Suggestions for Authors

The authors provided responses to all my previous concerns.